# FLARE: Finetuning ReLU And FIRE for Efficient Long-Context Inference

## Abstract

Deploying large language models (LLMs) on resource-constrained edge devices is challenging due to computational bottlenecks, memory bottlenecks, and – for long-contexts – specifically the Softmax operation in the attention mechanism. While using ReLU in place of Softmax has been explored, and FIRE as an alternative to RoPE has been explored for models trained from scratch, there has been little work towards exploring fine-tuning models to utilize these efficient algorithms, or the combination of the two.

In this paper, we contribute FLARE, a method for fusing Rectified Linear Activations (ReLU) with Relative Encodings (specifically FIRE), and we share a particular recipe which allows these to be fine-tuned effectively into existing models and fused to create efficient long-context inference. Following this recipe yields markedly better validation loss, long-context inference speed, and successfully introduces the property of length-generalization – the property where the model gains high accuracy for contexts lengths several times larger than trained – unlike RoPE – without further fine-tuning.

Once FIRE and ReLU are both fine-tuned into a model, we show these can be mathematically fused into a single, more efficient operation, which on average was found to shave 98.9% of FIRE operations and produce a Probability matrix with 98.9% zeros in its lower-triangle.

Finally, we benchmark inference speed improvements for custom hardware as well with custom CUDA kernels. Using Power, Performance, and Area (PPA) analysis, we show that FLARE operates at eight times the frequency of Softmax while consuming only 0.1% of the power and 0.11% of the energy per cycle. Our custom CUDA Kernel shows 3.8x faster operation than Softmax FlashAttention. We believe this shows the potential of fine-tuning new algorithms in pre-trained models, and we share our fine-tuning recipes, code and custom hardware designs at `https://anonymous.4open.science/r/nanoGPTBD54`.

## 1 Introduction

Transformers (Vaswani et al., 2017) have revolutionized natural language processing by enabling parallel processing of sequences through self-attention mechanisms. However, deploying transformer-based models on resource-constrained edge devices remains challenging, especially for long-context applications. Edge devices often lack the computational resources and memory bandwidth to efficiently implement latency-optimization techniques for Softmax, such as FlashAttention (Dao et al., 2022), resulting in significant performance bottlenecks. This problem has inspired alternative hardware algorithms for targeting custom hardware (Stevens et al. (2021), Liu et al. (2024)).

While using ReLU in place of Softmax is a computationally efficient alternative, it has been noted in literature that pre-training models with ReLU results in lower machine learning performance than Softmax (Wortsman et al. (2023), Shen et al. (2023), Zhang et al. (2021)). The potential efficiency improvements despite the performance gap has prompted numerous attempts to introduce additional operations and normalization techniques to mitigate the shortcomings of ReLU, including following the PV multiplication with a layernorm (Zhang et al., 2021) as well as specialized and learned divisors for the result of the ReLU activation (Wortsman et al. (2023), Shen et al. (2023)). Notably,

these prior had not explored fine-tuning ReLU into Softmax pre-trained Transformer models, or whether this could improve validation loss per token trained.

Context length mainly hinges upon the Transformer model's position encoding mechanism. While long-context operation via RoPE and the extension of context length with RoPE has been explored (Su et al. (2023), Chen et al. (2023)), the RoPE operation requires element-wise multiplications input vectors which, while not a bottleneck, are less computationally efficient than additive relative position encodings, such as FIRE Li et al. (2024).

In this paper, we address the hardware-efficient long-context operation. We make the following contributions:

- Experiments showing training with Softmax first and then fine-tuning ReLU as a direct elementwise replacement, yields significantly better next-token-prediction accuracy than training with ReLU from the start, for the same number of training steps and and input training tokens.
- The FLARE algorithm reorders adjacent FIRE and ReLU operations in inference, offering the ability to bypass addition operations in FIRE.
- A hardware implementation and performance, power, and area (PPA) analysis, showing substantial efficiency gains of ReLU over Softmax in specialized EdgeLLM-accelerators.
- A CUDA Kernel for accelerating a ReLU-Augmented forward-pass, with analysis of gains over Softmax-based FlashAttention, and PyTorch source code for supporting accelerated training and fine-tuning of FLARE and ReLU-Attention models.

## 2 BACKGROUND

Custom hardware for Transformer-based models must determine low-level implementations of self-attention mechanisms and positional encoding strategies, and hence are greatly affected by parallelism limitations and involved operations.

### 2.1 INEFFICIENCIES OF SOFTMAX IN SPECIALIZED HARDWARE

At the core of the Transformer is the self-attention mechanism, which relies on the Softmax function to compute attention scores. The Softmax function is defined as:

$$\text{Softmax}(x_i) = \frac{e^{x_i - s_{\max}}}{\sum_j e^{x_j - s_{\max}}} \tag{1}$$

where $s_{\max} = \max x_j$ is subtracted from each input to improve numerical stability.

### 2.2 SOFTMAX INEFFICIENCY IN HARDWARE

Implementing Softmax efficiently in hardware poses significant challenges:

1. **Non-linearity**: Implementing the exponential function in hardware requires approximations or lookup tables, adding need for larger silicon area.
2. **Sequential Operations**: The max and sum computations introduce data dependencies that limit parallelism, resulting in stalls in the pipelining.
3. **Long Context Inefficiencies**: Challenges in pipelining and the need to maintain precision for accumulated values – especially for the denominator – will inflate the size required for this module, with complexity scaling with max supported context length.

### 2.3 RELU AS A SOFTMAX REPLACEMENT

The Rectified Linear Unit (ReLU) function is defined as:

$$\text{ReLU}(x_i) = \max(x_i, 0) \tag{2}$$

Replacing Softmax with ReLU offers significant advantages when implementing in edge hardware:

- **Simplified Non-Linearity**: The total number of operations is $\frac{N^2}{2}$ comparisons, removing need for look up tables.

- **Element-wise Operation**: ReLU can be applied independently to each element, allowing full-parallelization and simplifying pipeline-implementations.

- **Long-Context Efficiency**: Via introducing full-parallelism, removing the normalization terms, and simplifying the non-linearity (essentially setting to zero if negative or value leaving as is), the total size of the resulting implementation of ReLU is not tied to the max context length.

## 2.4 Long-Context Scaling of Positional Embeddings

RoPE (Su et al., 2023) and Learned Absolute Position Embeddings (Vaswani et al., 2017) both tie a trained model a specific max supported context length. Should longer context length be desired, the Learned Absolute Position Embeddings table will become larger and linearly with the context length. RoPE would require fine-tuning, e.g. via interpolation Chen et al. (2023), at the longer-context length.

Functional Interpolation for Relative Position Encoding (FIRE) (Li et al., 2024) is a relative positional encoding method which results in additive bias after the $QK^T$ multiplication. This is a type of relative position encoding which exhibits context-length generalization. With FIRE (and many other additive embeddings) one can train on a smaller context length and then simply inference on a longer context lengths than trained. This encourages easy scalability of any hardware solutions, as well to longer context lengths.

## 3 Methods

In this section, we describe the methodologies employed in our study, including the experimental setup, datasets, training configurations, the detailed recipes for integrating ReLU and FIRE into Transformer models, the implementation of the FLARE algorithm, the development of the ReLU-augmented FlashAttention mechanism, and the hardware implementation and performance, power, and area (PPA) analysis.

### 3.1 Experimental Setup

Our experiments are conducted using PyTorch (Paszke et al., 2019) for model training and evaluation, and custom CUDA kernels for implementing and profiling ReLU-based attention mechanisms. We extend the NanoGPT repository (Karpathy, 2022) for language model fine-tuning and the llm.c repository (Karpathy, 2023) for implementing and benchmarking our ReLUFlashAttention kernels. The modified code and resources are available at the following anonymous links:[1][2].

All experiments are performed on NVIDIA A100 GPUs, leveraging their computational capabilities for efficient training and inference of large language models.

### 3.2 Dataset and Training Settings

We utilize the OpenWebText dataset (Gokaslan & Cohen, 2019), an open-source replication of the WebText corpus used for training GPT-2. The dataset consists of high-quality web content extracted from Reddit submissions with an upvote score of at least 3, providing a diverse and representative sample of internet text.

---

[1]https://anonymous.4open.science/r/nanoGPTBD54
[2]https://anonymous.4open.science/r/llmc27B0

For model training, we use a maximum context length of 1024 tokens. Unless otherwise specified, we initialize our models from the GPT-2 small checkpoint, which contains approximately 124 million parameters. Training is performed using a batch size of 32 and a learning rate of $6 \times 10^{-4}$, optimized with the Adam optimizer (Kingma & Ba, 2014) and a cosine learning rate decay schedule. Gradient clipping is applied with a maximum norm of 1.0 to stabilize training.

### 3.3 RECIPES FOR ReLU INTEGRATION

We conducted a series of experiments to compare different training recipes for effectively integrating the ReLU activation function as a replacement for the Softmax function in the attention mechanism of Transformer models. Our goal was to identify strategies that mitigate the performance degradation typically observed when training models with ReLU from scratch.

#### 3.3.1 EXPERIMENT 1: TRAINING FROM SCRATCH VS. FINE-TUNING WITH ReLU

In the first experiment, over the same total number of training iterations, we compared the performance of models trained with ReLU from scratch against models that were first trained with Softmax and then fine-tuned with ReLU.

- **Training with ReLU from Scratch**: We trained a Transformer model with the ReLU-based attention mechanism from scratch for 30,000 iterations. The model was initialized randomly and did not utilize any pre-trained weights. We tracked the validation loss at regular intervals to assess the learning progress.

- **Fine-tuning ReLU into a Softmax Pre-trained Model**: We first trained the model with the standard Softmax-based attention mechanism for 20,000 iterations. After this initial training phase, we replaced the Softmax function with ReLU and fine-tuned the model for an additional 10,000 iterations. We monitored the validation loss throughout both phases.

#### 3.3.2 EXPERIMENT 2: SEQUENTIAL VS. SIMULTANEOUS INTEGRATION OF ReLU AND FIRE

In the second experiment, we investigated the impact of integrating ReLU and FIRE positional embeddings into a pre-trained model, comparing sequential integration in different orders with simultaneous integration. Our goal was to determine the most effective strategy for incorporating both ReLU and FIRE and to understand how the order of integration affects model performance and training stability.

- **Simultaneous Training of ReLU and FIRE**: Starting from the GPT-2 124M checkpoint, we simultaneously replaced the Softmax function with ReLU in the attention mechanism and the absolute positional embeddings with FIRE. We then trained the model for 20,000 iterations, integrating both changes at once. Throughout the training process, we tracked the validation loss and monitored the input and output distributions of the ReLU activation function to observe how the model adapted to these simultaneous changes.

- **Sequential Training with ReLU Followed by FIRE**: We first fine-tuned the GPT-2 124M model by replacing the Softmax function with ReLU while retaining the absolute positional embeddings. We trained the model for 10,000 iterations to allow it to adapt to the ReLU activation in the attention mechanism. Subsequently, we replaced the absolute positional embeddings with FIRE and continued fine-tuning for another 10,000 iterations. During both phases, we monitored the validation loss and activation statistics to assess the model's adaptation and performance.

- **Sequential Training with FIRE Followed by ReLU**: Conversely, we also explored the effect of introducing FIRE before ReLU. Starting from the GPT-2 124M checkpoint, we replaced the absolute positional embeddings with FIRE while retaining the Softmax function and trained the model for 10,000 iterations to allow it to adapt to FIRE. Then, we replaced the Softmax function with ReLU and fine-tuned the model for an additional 10,000 iterations. We monitored validation loss and activation statistics throughout both training phases to evaluate the model's performance and stability.

By comparing these three integration strategies—simultaneous integration and sequential integration in both orders—we aimed to determine the most effective approach for incorporating both ReLU and FIRE into a pre-trained Transformer model. This comparison allowed us to assess how the order of integration influences the model's ability to adapt, the training stability, and the final performance in terms of validation loss and length generalization capabilities.

### 3.3.3 EXPERIMENT 3: LENGTH GENERALIZATION EVALUATION

In the final experiment, we assessed the ability of each trained model to generalize to longer context lengths than those used during training.

- **Evaluation on Extended Context Lengths**: We evaluated all models on context lengths of 2048 and 4096 tokens, to test length generalization from their trained maximum context length of 1024 tokens. We measured the validation loss at these extended lengths to determine how well each training recipe enabled length generalization.

This evaluation allowed us to understand the impact of different integration strategies on the models' ability to handle longer sequences, and acts as a figure of merit for how well FIRE was integrated.

### 3.4 IMPLEMENTATION OF THE FLARE ALGORITHM

To further improve computational efficiency, we proposed a fused FLARE algorithm which combines FIRE position encoding with sparsity producing ReLU. This algorithm reorders comparison of ReLU to occur before FIRE bias addition step for positional encoding.

In the standard attention mechanism with FIRE and ReLU, attention score matrix $\mathbf{S}$ is computed as:

$$s_{ij} = \max(a_{ij} + f_{ij}, 0) \tag{3}$$

where $\mathbf{A} = \mathbf{Q}\mathbf{K}^{\top}$ is the scaled dot-product of queries and keys, and $\mathbf{F}$ represents the FIRE bias matrix.

The FLARE algorithm optimizes this computation by observing that if $f_{ij} \leq -a_{ij}$, then $s_{ij} = 0$, and the addition can be skipped. This leads to a case-wise expression:

$$s_{ij} = \begin{cases} 0, & \text{if } f_{ij} \leq -a_{ij} \\ a_{ij} + f_{ij}, & \text{otherwise} \end{cases} \tag{4}$$

By implementing this logic, we can avoid unnecessary addition operations when $f_i j <= -a_i j$. We conduct experiments to measure the proportion of operations which take this branch, reporting in the results section.

Figure 1 illustrates the original attention architecture, the ReLU with FIRE, and the optimized FLARE module.

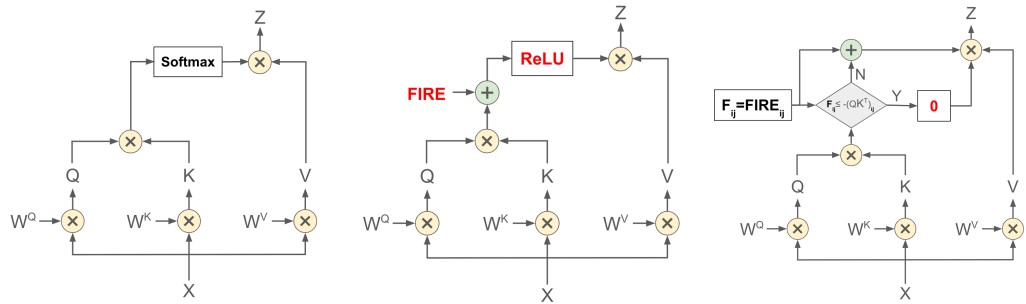

Figure 1: Original attention architecture, ReLU+FIRE, and FLARE

## 3.5 ReLU Attention CUDA Implementation

We contribute implementation and analysis and CUDA kernel for efficient forward pass of ReLU for facilitating community ML development of models utilizing using ReLU-attention.

### 3.5.1 Development of Custom CUDA Kernels

We developed custom CUDA kernels that implement the ReLU-based attention mechanism with tiling optimization techniques inspired by FlashAttention (Dao et al., 2022). By leveraging shared memory and tiling strategies, we optimized memory access patterns and reduced bandwidth requirements. This is crucial for handling large attention matrices, especially with long sequences.

The simplified ReLUFlashAttention forward pass algorithm is outlined in Algorithm 1.

---

**Algorithm 1** ReLUFlashAttention

---

**Require:** Matrices $Q, K, V \in \mathbb{R}^{N \times d}$ in high-bandwidth memory (HBM), on-chip shared memory (SM) of size $M$.
1: Set block sizes $B_c = \lfloor \frac{M}{4d} \rfloor$, $B_r = \min\left(\lfloor \frac{M}{4d} \rfloor, d\right)$.
2: Initialize $O = \mathbf{0}^{N \times d}$ in HBM.
3: Divide $Q$ into $T_r = \lceil \frac{N}{B_r} \rceil$ blocks $Q_1, \ldots, Q_{T_r}$ of size $B_r \times d$, and divide $K, V$ into $T_c = \lceil \frac{N}{B_c} \rceil$ blocks $K_1, \ldots, K_{T_c}$ and $V_1, \ldots, V_{T_c}$ of size $B_c \times d$.
4: **for** $j = 1$ **to** $T_c$ **do**
5:     Load $K_j, V_j$ from HBM to SM.
6:     **for** $i = 1$ **to** $T_r$ **do**
7:         Load $Q_i$ from HBM to SM.
8:         Compute $A_{ij} = Q_i K_j^\top$.
9:         Apply FLARE logic to compute $S_{ij}$:
10:             For each element $a_{pq}$ in $A_{ij}$ and $f_{pq}$ in $F_{ij}$:
11:             If $f_{pq} \leq -a_{pq}$, set $s_{pq} = 0$.
12:             Else, compute $s_{pq} = a_{pq} + f_{pq}$.
13:         Update $O_i \leftarrow O_i + S_{ij} V_j$.
14:     **end for**
15: **end for**
16: **return** $O$.

---

## 3.6 Hardware Implementation and PPA Analysis

To demonstrate the hardware efficiency gains of using ReLU algorithm over Softmax in custom hardware, we implemented these functions in a custom hardware and performed a performance, power, and area (PPA) analysis compared to a Softmax implementation from literature.

### 3.6.1 Hardware Synthesis

We synthesized hardware implementations of the ReLU function and the Softmax function using a 130nm CMOS process technology. The implementations were designed to:

- **Exploit Parallelism**: Both ReLU implementations were designed to operate on multiple inputs simultaneously, leveraging parallelism to increase throughput.
- **Minimize Critical Path Delays**: By simplifying the logic and avoiding complex operations, we reduced the critical path delays, allowing for higher operating frequencies.
- **Optimize Resource Utilization**: The designs aimed to minimize silicon area and power consumption, making them suitable for resource-constrained edge devices.

### 3.6.2 Performance Metrics

We measured key performance metrics for each implementation, including:

- **Maximum Operating Frequency (MHz)**: The highest frequency at which the circuit can operate reliably.
- **Power Consumption (mW)**: The total power consumed by the circuit during operation.
- **Energy per Cycle (pJ)**: The energy consumed per clock cycle.
- **Silicon Area (mm$^2$)**: The physical area occupied by the circuit on the chip.

### 3.6.3 ANALYSIS AND COMPARISON

By comparing the synthesized hardware modules, we quantified the efficiency gains of ReLU over Softmax in terms of speed, power efficiency, and hardware resource requirements. The results are detailed in Section 6.2.

## 4 RESULTS

### 4.1 PRE-TRAINING WITH SOFTMAX VS RELU

We start by comparing the performance of ReLU fine-tuning vs training from scratch.

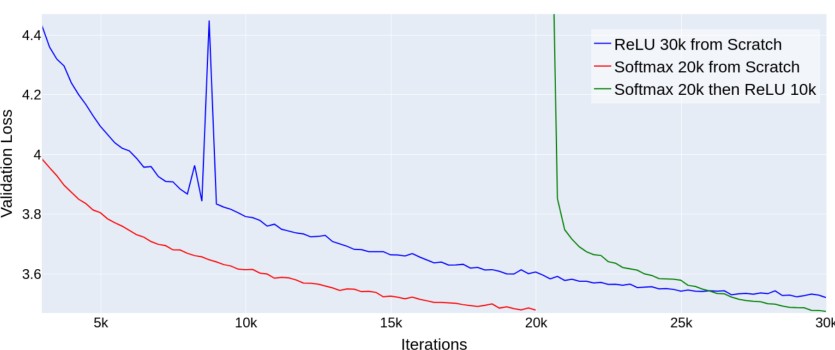

Figure 2: Comparison of validation loss between ReLU trained for 30k iterations vs fine-tuning for 10k iterations after training Softmax for 20k

Figure 2 shows that, over the same number of total training iterations, a model trained with Softmax and fine-tuned with ReLU performs better than a model trained using just ReLU. After replacing Softmax with ReLU, the validation loss quickly overtakes that of the model with ReLU from scratch.

Furthermore, due to benefits of highly optimized Softmax attention on GPU with FlashAttention, we observed all-in-all a reduction in training time for the fine-tuning recipe.

Training on a single A100 80GB GPU, the training from scratch with ReLU took 59% longer than the Softmax-Then-ReLU fine-tuning recipe. ReLU pre-training took 17 hours, where as the total training time from Softmax and ReLU fine-tuned took 12 hours in total.

### 4.2 RESULTS OF DIFFERENT RECIPES FOR FINE-TUNING RELU AND FIRE INTO MODELS

Between fine-tuning simultaneously with ReLU and FIRE, fine-tuning sequentially (first FIRE then ReLU or first ReLU then FIRE), the best validation losses are nearly the same with fine-tuning simultaneously and fine-tuning FIRE-then-ReLU.

However, we find definitely that only fine-tuning with FIRE first then ReLU was able to successfully impart length-generalization capability to the model, as evidenced form figure 5.

### 4.3 INPUT AND OUTPUT STATISTICS

We monitored input and output statistics during fine-tuning to understand how the model adapts its input and output ranges when accommodating a vastly different activation function in the attention module.

Figures 6 and 7 show that the output distributions for ReLU stabilize after around 3k iterations. While Softmax's distribution of inputs was very symmetrical, ReLU's distribution pulled towards the negative side, clearly selecting a smaller proportion of its outputs – on average only 1.1% of its outputs – for the non-zero values in the output probability matrix.

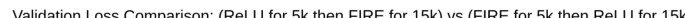

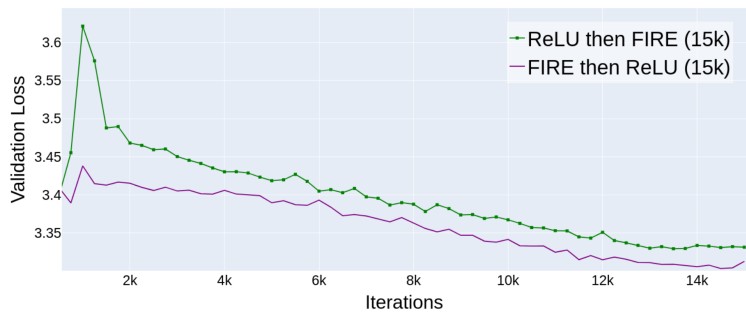

Figure 3: Comparison of fine-tuning GPT-2 models using either ReLU followed by FIRE or FIRE followed by ReLU. Above is shown only the last 15k of 20k iterations.

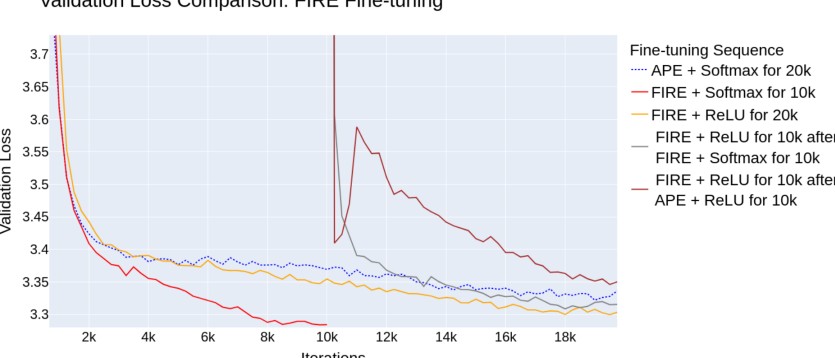

Figure 4: Validation loss for Softmax variants fine-tuned with FIRE

## 5 FLARE BRANCHING RESULTS

From evaluation of final checkpoint inference, we determined that the "Probability Matrix" output of ReLU is 98.9% zeros in the lower-left triangle for causal attention.

This high degree of sparsity means that that 98.9% of the time the $f_i j <= -a_i j$ branch will be taken, and the comparison alone will suffice for 98.9% of ReLU outputs.

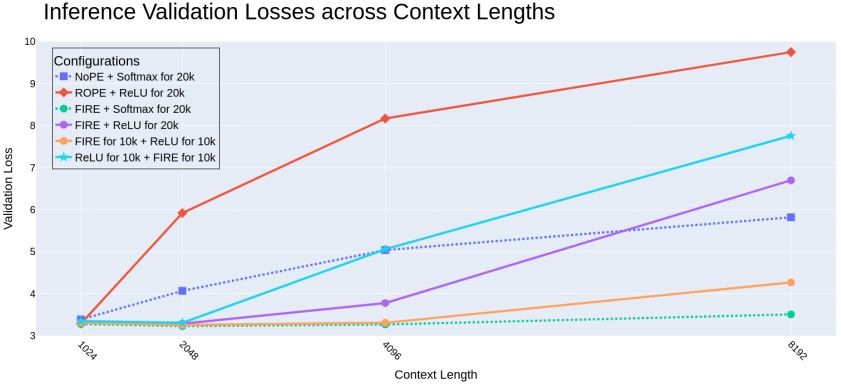

Figure 5: Validation loss measurements for each fine-tuning recipe, with NopE and RoPE provided as baselines. Note how fine-tuning Fire for 10k iterations then ReLU 10k iterations yields strong length-generalization vs other FIRE + ReLU recipes

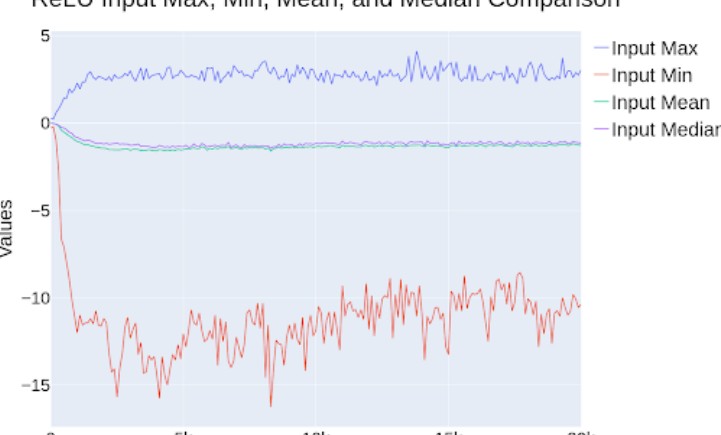

Figure 6: Input statistics of ReLU being finedtuned into a Softmax pre-trained model over 20k iterations

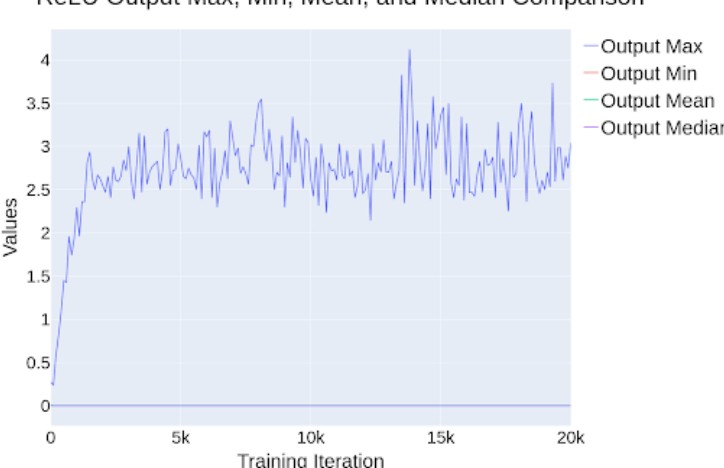

Figure 7: Output statistics of ReLU being fine-tuned into a Softmax pre-trained model over 20k iterations

## 6 PROFILING AND HARDWARE IMPLEMENTATION

### 6.1 PERFORMANCE EVALUATION

We compared the forward pass inference time between ReLUFlashAttention and FlashAttention across different context lengths. As shown in Figure 8, ReLUFlashAttention achieves an average speedup of 3.8x over FlashAttention for context lengths of 512 to 4096.

### 6.2 HARDWARE COMPARISON

We conducted a performance, power, and area (PPA) analysis between ReLU and Softmax implementations synthesized in a 130nm CMOS process. As summarized in Table 1, ReLU has:

- 8x higher maximum frequency.
- 0.1% of the power consumption of Softmax.
- 0.11% energy per cycle compared to Softmax.

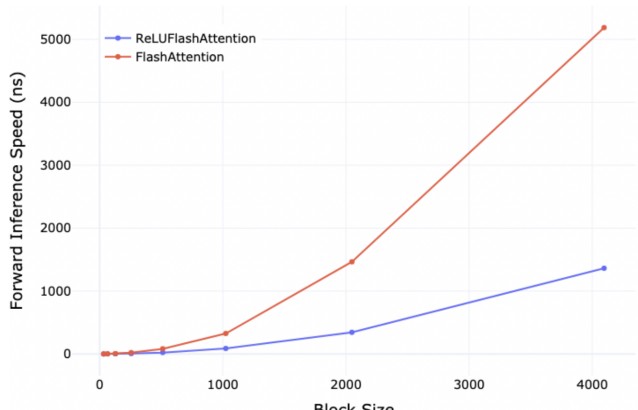

Figure 8: Inference time comparison between ReLUFlashAttention and FlashAttention

- Only 1% of the silicon area used by Softmax.

Table 1: Hardware comparison of ReLU and Softmax in 130nm CMOS

| Metric | ReLU | Softmax |
|---|---:|---:|
| Max Frequency (MHz) | 1667 | 200 |
| Area (mm$^2$) | 0.0028 | 0.2564 |
| Power (mW) | 4.87 | 516.28 |
| Energy per Cycle (pJ) | 2.9 | 2581.4 |

## 7 CONCLUSION

In this paper, we explore fine-tuning as a method for improving performance ReLU in the attention block for Transfomers. We demonstrate that fine-tuning models with ReLU achieves better validation-loss-per-iteration with ReLU from scratch, without adding additional compute steps.

We also show that with the correct fine-tuning-recipe, one can introduce both FIRE and ReLU into a pre-trained model, obtaining the length-generalization, better improved final validation loss, and high (98.9%) zeros introduced via ReLU's output sparsity.

Our proposed FLARE algorithm efficiently fuses ReLU and FIRE, significantly reducing computational complexity by reordering the computation.

Finally, our analysis shows that ReLU offers substantial performance, power, and area advantages over Softmax, making it highly suitable for edge devices. The ReLU-augmented FlashAttention implementation can catalyze training of theser efficient models.

Overall, we hope this work relays the possibilities within exploration of fine-tuning recipes for various efficient algorithms, and support unblocking deploying hardware-friendly large language models on edge devices without compromising performance or longer-context capabilities.

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
