# OpenReview forum: "FLARE: Fine-tuned Long-context Acceleration with ReLU-enhanced FIRE"
_ICLR.cc/2025/Conference — Submitted to ICLR 2025_

### Official Review · Reviewer_XWKy · 2024-10-21

**Soundness:** 2
**Presentation:** 2
**Contribution:** 2
**Rating:** 5
**Confidence:** 4

**Summary:**

Deploying transformer models on edge devices with limited battery life presents significant challenges, particularly for long-context applications. The Softmax operation often becomes a major bottleneck due to constraints like reduced memory bandwidth and parallelism, making it hard to utilize latency-optimizing techniques. This paper proposes a solution by fine-tuning the ReLU function as a replacement for Softmax, along with utilizing Functional Interpolation for Relative Position Encoding (FIRE), which improves model efficiency and maintains accuracy. The resulting algorithm, ReLU-enhanced FIRE (FLARE), combines these techniques, reducing power consumption and operational complexity, which is particularly suitable for large language models on edge devices.

**Strengths:**

1.	The tackled problem is relevant to the community.
2.	The proposed method is useful for reducing the computation complexity of large language models.

**Weaknesses:**

1.	The limitations of the related work and how these issues are addressed in the proposed method should be clarified.
2.	This work builds upon several related works and combines multiple techniques. A clear distinction between what is novel in the proposed method and what is implemented based on existing methods is required.
3.	The proposed method should be discussed in more detail describing all the operations involved. Please also explain all the design decisions made to develop it.
4.	The experimental setup and tool flow used to conduct the experiments should be discussed in more detail.

**Questions:**

1.	What are the limitations of this paper and what are the potential impacts at large scale of this work?

---

> ### Author Response · Authors · 2024-11-28
> **Thank you for your feedback**
>
> >The limitations of the related work and how these issues are addressed in the proposed method should be clarified.
>
> Thanks again for your feedback, in our revision we worked to clarify the limitations of the related work and to clarify our contribution within this research area, revising our introduction to discuss both relations to prior work and our contributions.
>
> >This work builds upon several related works and combines multiple techniques. A clear distinction between what is novel in the proposed method and what is implemented based on existing methods is required.
>
> Indeed, we have worked to clarify this in our latest revision, and, in summary, we aim to determine method to improve the performance of ReLU based attention, but unlike prior work, instead of doing so via adding operations or hyperparameters, in order to achieve maximal hardware efficiency, we do not modify the ReLU attention, but instead utilize Softmax to pre-train and show we can anneal the Softmax model into an efficient ReLU model via finetuning.
>
> We show that this results in a better final language modeling capability with the same total number of tokens input for training.
>
> >The proposed method should be discussed in more detail describing all the operations involved. Please also explain all the design decisions made to develop it.
> >The experimental setup and tool flow used to conduct the experiments should be discussed in more detail.
>
> Thanks for your feedback, this discussions is now added within our latest revision.
>
> Questions:
> What are the limitations of this paper and what are the potential impacts at large scale of this work?
>
> Indeed, the techniques of finetuning ReLU, and demonstration that permutation space of finetuning sequence needs to be explored for optimizing performance while doing algorithmic finetuning, have potentially far reaching implications, even for larger models.
>
> Due to the vastly different output range of ReLU vs Softmax, we did not expect finetuning ReLU in would work even better than training with ReLU from scratch.
>
> By sharing this we hope to inspire even more daring finetuning adaptation attempts and finetuning sequence explorations -- especially those adaptations like ReLU which can significantly increase the final model's sparsity and hardware efficiency.

---

### Official Review · Reviewer_3d4a · 2024-10-30

**Soundness:** 2
**Presentation:** 4
**Contribution:** 2
**Rating:** 5
**Confidence:** 3

**Summary:**

The paper tackles the challenge of deploying large language models (LLMs) on edge devices by optimizing attention mechanisms. Specifically, the authors replace the Softmax operation used in traditional transformers with ReLU, aiming to improve computational efficiency for long-context sequences. They also integrate FIRE (Functional Interpolation for Relative Encoding) to improve the handling of long input sequences. The paper introduces a system named FLARE, which combines ReLU-based attention with FIRE encoding and fine-tunes it on GPT-2 to showcase the performance benefits in terms of speed, efficiency, and memory usage.

**Strengths:**

While the idea of using ReLU in place of Softmax has appeared in prior research, this paper is novel in the sense that it combines it with FIRE positional encoding for long-context acceleration. The focus on deploying these techniques for edge devices is timely and practically relevant, given the rising demand for efficient LLMs in constrained environments.

Besides, the paper presents a well-structured experimental evaluation, profiling memory usage, speed, and power efficiency. The experiments are thoughtfully designed to show how ReLU-based attention performs compared to Softmax on long-context inference.

In terms of the writing quality, the paper is clear and easy to follow, with well-defined objectives and explanations of both ReLU and FIRE techniques. The figures and tables help convey the improvements in speed and power usage, though additional comparisons with other positional encodings would have been helpful.

**Weaknesses:**

1. The core ideas—using ReLU instead of Softmax and FIRE for positional encoding—are borrowed from prior work. The contribution lies mainly in engineering and integration, rather than in proposing a new method or theory. This makes the paper more of an “A + B”-style contribution.

2. The paper primarily evaluates performance on GPT-2. It is unclear how well the proposed optimization generalizes to larger models like Qwen, LLaMA, or LLaMA-2. Could these gains be replicated on models with billions of parameters? Further benchmarking would have strengthened the paper. As long as the authors mentioned that they had access to 80GB A100s, I think larger and more-updated models shall be fine-tuned in the similar approach with FSDP support, and the results shall be included in the experiments to demonstrate the effectiveness of the purposed algorithms,.

3. The paper claims the target of improving inference efficiency on edge devices, but only the experiment of PPA hardware has been provided. In real edge devices like an android device, the power measurement is not that straightforward. I know the authors could not directly use edge-LLM implementations like Llama.cpp to test on a real edge because modifications on the backend and the computation-graph formulation stage are required, but I think it is fine to implement a single attention block and test on different real edge platforms, to give a more direct sensing of the effectiveness of the proposed algorithms.

4. I think the only validation loss could not demonstrate the ability of a language model before and after fine-tuning. More down-stream tasks are more important than validation loss.

**Questions:**

My question is summarized as:

1.  Can the proposed method scale to larger models like Qwen, LLaMA, or LLaMA-2?

2.  How would the method perform on real edge devices, like Android phones? Could a single attention block be tested on such devices for more practical insights?

3. Why were downstream tasks not included to assess the fine-tuned model’s real-world performance beyond validation loss?

---

> ### Author Response · Authors · 2024-11-28
> **Thank you for your feedback**
>
> > The core ideas—using ReLU instead of Softmax and FIRE for positional encoding—are borrowed from prior work. The contribution lies mainly in engineering and integration, rather than in proposing a new method or theory. This makes the paper more of an “A + B”-style contribution.
> Thanks for your feedback, after reading the reviews, we realized we really needed to clarify and explain our main contributions, adding direct enumeration of main contributions and relation to prior work to our revision in the introduction section.
>
> One specific contribution I must clarify, is around our contribution with respect to improving ReLU performance, without detracting from its HW Efficiency:
> While other works have focused on augmenting ReLU with additional operations and hyperparameters to improve its perplexity when language modeling, adding operations would complicate the ultimate hardware implementation.
> With the goal of targeting maximum HW Efficiency, we explore method that allows to significantly boost ReLU's ML performance without adding operations.  Without any additions to the algorithm, we are able to improve both the validation loss per token trained and obtain optimal PPA for our implementation in hardware.
>
> > The paper primarily evaluates performance on GPT-2. It is unclear how well the proposed optimization generalizes to larger models like Qwen, LLaMA, or LLaMA-2. Could these gains be replicated on models with billions of parameters? Further benchmarking would have strengthened the paper. As long as the authors mentioned that they had access to 80GB A100s, I think larger and more-updated models shall be fine-tuned in the similar approach with FSDP support, and the results shall be included in the experiments to demonstrate the effectiveness of the purposed algorithms,.
>
> Thanks for the feedback, we worked to clarify the reasons for the scope of our paper.
>
> Our main emphasis within this paper is building efficient models targeting custom EdgeLLM accelerator hardware.
>
> Currently, the size of models which run efficiently on edge hardware is in the hundreds of millions of parameters, mainly due to Edge Memory Limitations and high energy costs of off-chip DRAM accesses [1][2][3].
> https://arxiv.org/pdf/2402.14905
> https://arxiv.org/pdf/2403.12844
> https://arxiv.org/pdf/2402.16840
>
> And so improving both perplexity and hardware efficiency of our custom hardware for this models in this range is our group's focus.
>
> >The paper claims the target of improving inference efficiency on edge devices, but only the experiment of PPA hardware has been provided. In real edge devices like an android device, the power measurement is not that straightforward. I know the authors could not directly use edge-LLM implementations like Llama.cpp to test on a real edge because modifications on the backend and the computation-graph formulation stage are required, but I think it is fine to implement a single attention block and test on different real edge platforms, to give a more direct sensing of the effectiveness of the proposed algorithms.
> I think the only validation loss could not demonstrate the ability of a language model before and after fine-tuning. More down-stream tasks are more important than validation loss.
>
> We definitely hear that end-to-end figures would bolster the paper for overall estimates of model inference speed, for custom hardware this would require depend on design of the whole of the LLM Accelerator, and I hope it may not too much for me to suggest this may be out of scope for this particular paper.
>
> That being said, in the realm of custom hardware the softmax bottleneck has been well documented, spurring design of more hardware friendly implementations, citing TTFT equivalent bottlenecks in long context, namely https://arxiv.org/abs/2103.09301
>
>
> > Can the proposed method scale to larger models like Qwen, LLaMA, or LLaMA-2?
> While not the main focus of this work, we highly suspect it could be, though while smaller models tend to be more robust to large changes, we would likely need to more smoothly transition between the two representations to allow the larger model more time to adjust
>
> > How would the method perform on real edge devices, like Android phones? Could a single attention block be tested on such devices for more practical insights?
>
> While mentioned this isn't the target area of our work, this is interesting. Due to the pipelining effects of converting to fully-parallelized ReLU operations, the line would be blurry where attention itself starts and stops if doing a single module, but we can likely derive an end-to-end estimate in this manner, in so far as we can find a compiler that can utilize the extremely large sparsity opportunities produced.
>
> > Why were downstream tasks not included to assess the fine-tuned model’s real-world performance beyond validation loss?
> Thank again the feedback -- we will certainly pursue adding this into our repository.

---

### Official Review · Reviewer_DCxp · 2024-11-02

**Soundness:** 2
**Presentation:** 3
**Contribution:** 2
**Rating:** 3
**Confidence:** 3

**Summary:**

The paper presents the ideas to improve the performance and energy consumption for LLM models. Ideas presented in the paper can be summarized in the following points
* Replacing Softmax with ReLU for attention blocks.
* Integrating the above change in FIRE positional encodings to show the effectiveness of the technique for a long context (termed as FLARE).
* Study of improvement in performance of softmax block on GPU.
* Study of improvement in PPA of hardware implementation of FLARE vs Softmax.

**Strengths:**

* Paper has been written clearly with progressive introduction of concepts.
* It captures and presents interesting insights about the training models with FLARE and Softmax.

**Weaknesses:**

* The major weakness is the lack of novelty (or a lack of clarity in conveying the primary contribution). To the best of my understanding, the paper provides the following contributions:

  * Replacing Softmax with ReLU for attention blocks.
  > * The use of ReLU instead of Softmax is well-studied (e.g., [1] [2]). The paper provides evidence of the feasibility of replacing Softmax with ReLU; however, the results align with expectations set by [1]. As noted in reference [1], "we observe that attention with ReLU divided by sequence length can approach or match traditional Softmax attention in terms of scaling behavior as a function of compute for vision transformers." This is consistent with the results presented in the paper. It will be helpful to prominently mention any new insights that will be helpful for the community.
  * Integrating the above change in FIRE positional encodings to show the effectiveness of the technique for a long context (termed as FLARE).
  > * Line 260-261: "At time of writing, we haven’t seen any previous attempts to finetune FIRE encodings into models to add longer context capabilities". Ref [3], the paper which introduced FIRE contains the section named "FINETUNING ON LONG TEXT BENCHMARK". Please highlight the difference with your work.

  * Study of performance improvement in the softmax block on GPU.
  > The practical enhancement achieved by using ReLU with zero-skipping is impressive for both CUDA and hardware implementation. However, an analysis of the overall impact on the model's execution time is lacking. Please include the end-to-end execution time of the models presented in the paper, as this will help underscore the comprehensive impact of the proposed change.

  * Study of improvements in PPA for hardware implementation of FLARE vs. Softmax.
  > The significance of this study would be strengthened by adding the percentage area of the blocks in a DNN ASIC accelerator or alternative metrics to quantify the overall impact of these hardware modifications.

[1] Replacing softmax with ReLU in Vision Transformers, https://arxiv.org/abs/2309.08586v2
[2] A Study on ReLU and Softmax in Transformer https://arxiv.org/abs/2302.06461
[3] Functional Interpolation for Relative Positions Improves Long Context Transformers, https://arxiv.org/abs/2310.04418

**Questions:**

* Is there a reason for using the 130nm PDK for the PPA comparison, given that it is several generations old? The results could differ significantly on more recent technologies, and employing a newer technology node would enhance the credibility of the results.
> Some more advanced open-source PDKs are available, such as [1][2]. However, depending on the tools used, integration may not be straightforward.

[1] https://github.com/mflowgen/freepdk-45nm
[2] https://eda.ncsu.edu/freepdk15/

---

> ### Author Response · Authors · 2024-11-28
> **Thank you for your feedback**
>
> Thank you for your feedback.
>
> We worked to revised the paper with your feedback in mind, creating a section to clarify our report of contributions.
> The main contributions of the paper are now added and itemized clearly within the introduction section.
>
> > Replacing Softmax with ReLU for attention blocks.
> The use of ReLU instead of Softmax is well-studied (e.g., [1] [2]). The paper provides evidence of the feasibility of replacing Softmax with ReLU; however, the results align with expectations set by [1]. As noted in reference [1], "we observe that attention with ReLU divided by sequence length can approach or match traditional Softmax attention in terms of scaling behavior as a function of compute for vision transformers." This is consistent with the results presented in the paper. It will be helpful to prominently mention any new insights that will be helpful for the community.
>
> Absolutely, in our revision we've worked to clarify our contribution here.  We demonstrate that with Finetuning, for the same amount of training tokens, we can obtain better final model next token prediction accuracy _without_ having to add learned hyperparameters or additional normalization steps. With this approach, we are able to obtain better performance per token trained and per second of training, while not modifying ReLU so that we can have it's full hardware efficiency advantages.
>
> > Line 260-261: "At time of writing, we haven’t seen any previous attempts to finetune FIRE encodings into models to add longer context capabilities". Ref [3], the paper which introduced FIRE contains the section named "FINETUNING ON LONG TEXT BENCHMARK". Please highlight the difference with your work.
>
> Yes, the above is from the FIRE paper; however, Tteir reported finetuning does not change the model architecture, but instead finetunes their pre-existing architecture, which already contains FIRE embeddings, towards a particular benchmark.
>
> Also, amongst the citations of FIRE paper we have not encountered replacement of RoPE or other embeddings with FIRE via finetuning, and so we believe ours is the first report of success in literature of finetuning specifically FIRE in place of the existing position embedding, showing FIRE's length-generalization can be quickly added to a pre-trained model.
>
>
> >Study of performance improvement in the softmax block on GPU.
> The practical enhancement achieved by using ReLU with zero-skipping is impressive for both CUDA and hardware implementation. However, an analysis of the overall impact on the model's execution time is lacking. Please include the end-to-end execution time of the models presented in the paper, as this will help underscore the comprehensive impact of the proposed change.
>
> Thank you for this great insight, this is very true, and while we were not able to fit into this revision, we will certainly pursue adding statistics end-to-end detailed profiling on our repository.
>
> > Study of improvements in PPA for hardware implementation of FLARE vs. Softmax.
> The significance of this study would be strengthened by adding the percentage area of the blocks in a DNN ASIC accelerator or alternative metrics to quantify the overall impact of these hardware modifications.
>
> Again, excellent feedback, in our latest feedback we attempted to clarify this qualitatively, and will work with our EDA tools to see if we can determine deltas from progressive conversion of Softmax into the FLARE hardware implementation. While this will require greater analysis than for this iteration, we will certainly pursue this, and very much appreciate the feedback.
>
>
> >Is there a reason for using the 130nm PDK for the PPA comparison, given that it is several generations old? The results could differ significantly on more recent technologies, and employing a newer technology node would enhance the credibility of the results.
> Some more advanced open-source PDKs are available, such as [1][2]. However, depending on the tools used, integration may not be straightforward.
>
> Good question, and I will share a few of our reasons for opting for the 130nm PDK:
> 1. FreePDK45 while 45nm, is not manufacturable or based on a real process. (from the Github Page: "The FreePDK45 kit is an open-source generic process design kit (PDK) (i.e., does not correspond to any real process and cannot be fabricated")
> 2. While Skywater 130nm is not as advanced a node, it is both open-source and manufacturable, utilized widely, is vetted by a large number taped out and successful ASICs, and since it is open-source and vetted in real silicon, is often used for sharing relative PPA advantages for different approaches for publications.
>
> Thanks again, we worked hard to incorporate your feedback into our revision.

---

### Official Review · Reviewer_TXyv · 2024-11-04

**Soundness:** 2
**Presentation:** 1
**Contribution:** 3
**Rating:** 3
**Confidence:** 4

**Summary:**

This paper propose to combine FIRE position encoding method with ReLU activation to serve as an efficient attention mechanism for long context input. It evaluates on GPT-2 within a commercial EDA tool. The experiment is designed to fine-tune GPT-2 with the new architecture on OpenWebText data. The validation loss and inference speed is compared to demonstrate the effectiveness of the method.

**Strengths:**

1. reasonable but less common paper writing flow.
2. effective proposal on combining FIRE with ReLU to replace softmax based attention.
3. solid implementation on proposed ReLUFlashAttention and hardware profiling.

**Weaknesses:**

1. the paper presentation is poor, e.g., typos, figure formatting, section organization, etc.
2. the idea is driven by single model experiment and lack of insightful analysis.

**Questions:**

1. L83-L90, the notation "n" is missing definition.
2. L322, "don't" is less formal.
3. Fig. 8 and Fig. 9, captions are not centered.
4. The number 20K seems to be model dependent, and it cannot generalize to other models. There is no guarantee how long to fine-tune the proposed method, which is also mentioned as a concern of instability from using ReLU as in L151.
5. why only compare ReLU and softmax in section 8?
6. where are other curves in figure 3, 5 and 7?

---

> ### Author Response · Authors · 2024-11-28
> **Thank you for your feedback**
>
> Thank you for your feedback, we removed typos from the paper, centered captions, and explored experiments for clarifying remaining questions.

---

### Meta-Review · Area_Chair_q4Wj · 2024-12-20

**Metareview:**

This work proposes using ReLU and FIRE to improve efficiency on edge devices, offering good power savings and speed-ups. While the integration is neat, the novelty is limited since it mostly reuses existing techniques. The experiments focus only on GPT-2 and lack downstream tasks or real device benchmarks. Though the authors clarified some points, key concerns remain. They must show how this scales, test bigger models, and provide more thorough evaluations. Without stronger evidence of impact and novelty, this work is not ready to be accepted.

**Additional Comments On Reviewer Discussion:**

Reviewers raised concerns about: 1) novelty of combining ReLU with FIRE, viewing it as an "A+B" contribution, 2) limited evaluation on only GPT-2 rather than larger models, 3) lack of downstream task evaluation beyond validation loss, and 4) insufficient edge device testing. Authors addressed these by clarifying their novel contributions in improving ReLU performance without additional operations, explaining their focus on edge-suitable model sizes, and committing to add downstream evaluations to their repository. The authors engaged constructively with all feedback, though final ratings remained unchanged, ranging from reject (3) to marginally below acceptance threshold (5).

---

### Decision · Program_Chairs · 2025-01-22

Reject